

# In orbit cross-calibration of millimeter conically scanning spaceborne radars

Alessandro Battaglia[1,2,3], Filippo Emilio Scarsi[1], Kamil Mroz[3], and Anthony Illingworth[4]

[1]Politecnico of Torino, Italy
[2]University of Leicester, UK
[3]National Centre for Earth Observation, UK
[4]University of Reading, Reading, UK

**Correspondence:** Alessandro Battaglia
alessandro_battaglia@polito.it

**Abstract.** The planned and potential introduction in the global satellite observing systems of conically scanning Ka and W band atmospheric radars [e.g. the radars in the Tomorrow.IO constellation, https://www.tomorrow.io/space/, and the Wivern (WInd VElocity Radar Nephoscope) radar, www.wivern.polito.it] calls for the development of methodologies for calibrating and cross-calibrating these systems. Traditional calibration techniques pointing at the sea surface at about 12° incidence angle
are in fact unfeasible for such fast rotating systems.

This study proposes a cross-calibration method for conically scanning spaceborne radars based on ice cloud reflectivity probability distribution functions (PDF) provided by reference radars like the GPM Ka-band radar or the W-band radars planned for the ESA-JAXA EarthCARE or for the NASA Atmosphere Observing System missions. In order to establish the accuracy of the methodology, radar antenna boresight positions are propagated based on four configurations of expected satellite orbits
so that the ground-track intersections can be calculated for different intersection criteria, defined by cross-over instrument footprints within a certain time and a given distance. The climatology of the calibrating clouds, derived from the W-band CloudSat and Ka-band GPM reflectivity records, can be used to compute the number and the spatial distribution of calibration points. Finally, the mean number of days required to achieve a given calibration accuracy is computed based on the number of calibration points needed to distinguish a biased reflectivity PDF from the sampling-induced noisiness of the reflectivity PDF
itself.

Findings demonstrate that it will be possible to cross-calibrate within 1 dB a Ka-band (W-band) conically scanning radar like that envisaged for the Tomorrow.io constellation (Wivern mission) every few days (a week). Such uncertainties are generally meeting the mission requirements and the standards currently achieved with absolute calibration accuracy.





## 1 Introduction

Recent studies and advances in technology have given a great spur to the development and design of Earth observation missions involving rapidly conically scanning millimeter cloud and precipitation radars. Specifically Tomorrow.IO, a US private company, is currently building a constellation of miniaturized Ka-band (35 GHz) wide-swath conically and cross-track scanning radars with the goal of providing global coverage of precipitation with temporal resolution needed for operational applications
(i.e. with an average revisit time of about one hour for any given location). This novel observing system will enable more accurate forecasts of precipitation and extreme weather events to help people, countries and businesses mitigate the impact of severe weather events expected to exacerbate in a warming climate. A W-band conically scanning radar with polarization diversity Doppler capabilities aimed at providing in-cloud winds for improving numerical weather prediction is also undergoing ESA Earth Explorer 11 selection program Phase 0 studies as part of the Wivern mission (Illingworth et al. (2018); Battaglia
et al. (2018, 2022)).

The conically scanning strategy has the clear advantage of sampling larger domains and of reducing the effect of the clutter (Meneghini and Kozu, 1990; Illingworth et al., 2020). However, it makes the standard external calibration procedure, used for the CloudSat CPR (Tanelli et al., 2008), for several airborne instruments (Li et al., 2005; Battaglia et al., 2017; Wolde et al., 2019; Ewald et al., 2019) and planned for the EarthCARE radar (Illingworth et al., 2015), impractical since it requires
the antenna to be pointed at the ocean surface at an incidence angle of 10 degrees, a condition for which the ocean surface normalised backscattering cross section is insensitive to changes of the wind speed and the wind direction. This requires alternative calibration methods for the upcoming conically scanning space-borne radars. The use of natural targets (mainly rain) has been proposed for the calibration of (polarimetric) ground-based millimeter radar systems (Hogan et al., 2003; Myagkov et al., 2020) but is unfeasible from space because of the presence of attenuation, difficult to be accounted for, and of the
absence of spectral polarimetric observations. Alternatively (Protat et al., 2011; Kollias et al., 2019), ground-based mm-radars have been cross calibrated with the space-borne reference provided by the CloudSat CPR (whose calibration is believed to be accurate to within 0.5–1 dB). The idea initially proposed by Protat et al. (2009) is to compare mean vertical profiles of nonprecipitating ice cloud radar reflectivity after both the CPR and the ground-based radars have been degraded to the same sensitivity. Non precipitating ice clouds are selected because for such clouds it is possible to compute the effective reflectivities
from both observation points by simply correcting for gas attenuation. The mean vertical profiles are built based on contour frequency altitude displays collected from the ground-based radars within a given time relative to the satellite overpass (of the order of ±1h) and within a certain distance from the site (typically 200-300 km) for the CloudSat data.

In this work a similar rationale is followed, with the conically scanning radars being calibrated by other space-borne radars that are routinely calibrated with the standard ocean surface return procedure. For the Ka-band the GPM-DPR, expected to fly
till the end of the decade (Skofronick-Jackson et al., 2016; Battaglia et al., 2020), represents a solid choice for the reference calibrator thanks to internal and external calibration procedures that reach an accuracy better than 1 dB (Masaki et al., 2022) whereas the EarthCARE CPR and, later in this decade, the radars envisaged to be part of the NASA Atmosphere Observing System (AOS) constellation (Kollias et al., 2022) should provide a viable option of well calibrated W-band radars (within





1 dB as well) obtained via the ocean surface calibration method. The key science question underpinning this work is: what
cross-calibration accuracy can be achieved when intercalibrating the conically scanning and the reference radars in a given
time period? We will define a widely applicable approach to address this science question. This cross-calibration methodology
is described in Sect. 2. The technique is applied to four different configurations of orbit intersections. Results and expected
perfomrmances are presented in Sect. 3 with summary and discussions in Sect.4.

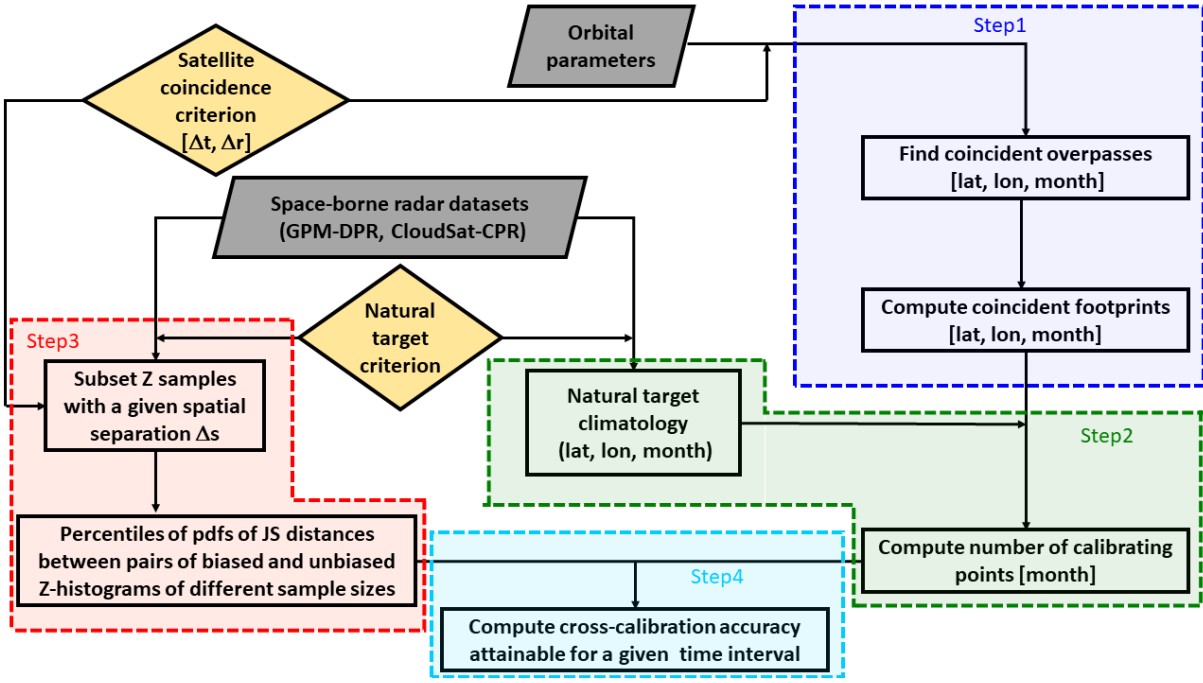

**Figure 1.** A flow chart showing the methodology followed for the cross-calibration between different spaceborne sensors.

## 2 Methodology for cross-calibration

The general methodology used to cross-calibrate different space borne radar systems is illustrated in the flow chart of Fig. 1.
The assumption underpinning the whole procedure is that there is a pair of satellites (one of which is the reference calibrator)
whose orbits have a sensible number of intersections. There are four steps needed for assessing the accuracy of the cross-
calibration.

**Step 1.** Once a satellite coincidence criterion is defined (observations are "coincident" if they are within a certain time
window $\Delta t$ and a certain distance $\Delta r$) the orbits of the two satellites are computed via the orbital parameters; the observing
geometry of the two systems is then used to compute the "coincident" footprints of the two radars. This is generally a strong
function of the latitude and a weak function of the longitude and the time of the year.



**Step 2.** Once the definition of the cross-calibrating targets has been established, a climatology of the mean number of layers for a given location and for each month is computed based on auxiliary data of past or existing missions employing mm-radars.

The thickness of the layers is determined by the coarser vertical resolution of the two radars that must be cross-calibrated. This climatology is then combined with the number of coincident footprints and allow to compute the mean number of calibrating points per unit time (e.g. weekly).

**Step 3.** The dataset of historical spaceborne mm-radar measurements can also be exploited to group reflectivity data into sample pairs separated by a given separation, $\Delta s$. The selection of $\Delta s$ is driven by the initial satellite coincidence criterion

with a conversion:

$$\Delta s = \sqrt{\Delta r^2 + v_{wind}^2 \Delta t^2} \qquad (1)$$

where $v_{wind}$ is the mean value of the wind speed moving the calibrating natural targets. For each pair of samples for a given separation and with a given number of samples, probability distribution functions (PDFs) of reflectivity are built. In order to quantify the similarity between two PDFs, $P$ and $Q$, the Jensen-Shannon (JS) distance is used (Endres and Schindelin, 2003).

It is defined as:

$$d_{JS} = \sqrt{\frac{\mathcal{D}_{KL}(P,M) + \mathcal{D}_{KL}(Q,M)}{2}} \qquad (2)$$

where $M = (P+Q)/2$ and $\mathcal{D}_{KL}$ is the Kullback–Leibler divergence defined as:

$$\mathcal{D}_{KL}(P,Q) = \sum_x P(x) \log_2 \frac{P(x)}{Q(x)}. \qquad (3)$$

A large ensemble of reflectivity (Z) PDFs is constructed for evaluation of the mean behaviour and the variability of JS

distances between Z-PDFs depending on the given amount of calibration data. A vast range of calibration calibration points from few thousands to millions has been considered. This allows to establish what is the statistical noise in the distance between PDFs when drawing a sample from ice calibrating clouds that are separated by a given distance, which mimics the process of cross-calibration.The impact on the JS distance when biasing one of the two PDFs by different miscalibration constants can also be established.

**Step 4.** As a result of step 3, it is possible, for any satellite coincidence criterion, to assess what calibration bias will be discernible for a given sample size. Then, via the results of step 2, the time needed to collect this number of calibrating points can be computed. By repeating the analysis for different satellite coincidence criteria, the optimal calibration procedure can be found.

## 2.1 Orbit intersections (step 1)

First, we want to establish how many coincidence points can actually be achieved between conically scanning radar systems orbiting in polar or inclined orbits and the reference radars. It is very unlikely that two different radars sensors on different orbits could illuminate the same target at the same time. Therefore a looser coincidence criterion is defined by assuming two observations from different platforms to be coincident if they are taken within a certain time interval, $\Delta t$, and a certain distance,





**Table 1.** Different combinations of temporal and spatial constraints considered in this study to define a "coincidence". The third column has been computed by using Eq. 1 with $v_{wind} = 20 \, \mathrm{ms}^{-1}$ which is a sensible value for upper level winds.

| Criterion # | Time constraint $\Delta t$ [minutes] | Distance constraint $\Delta r$ [km] | $\Delta s$ [km] |
|:---:|:---:|:---:|:---:|
| 1 | 15 | 100 | 101.6 |
| 2 | 15 | 200 | 200.8 |
| 3 | 15 | 500 | 500.3 |
| 4 | 15 | 1000 | 1000.2 |
| 5 | 15 | 2000 | 2000.1 |
| 6 | 30 | 100 | 106.3 |
| 7 | 30 | 200 | 203.2 |
| 8 | 30 | 500 | 501.3 |
| 9 | 30 | 1000 | 1000.6 |
| 10 | 30 | 2000 | 2000.3 |
| 11 | 45 | 100 | 113.6 |
| 12 | 45 | 200 | 207.2 |
| 13 | 45 | 500 | 502.9 |
| 14 | 45 | 1000 | 1001.5 |
| 15 | 45 | 2000 | 2000.7 |

$\Delta r$, from each other. Different combinations of temporal and spatial constraints adopted in this study are shown in Tab. 1. Goal
of the next investigation is to determine how the coincidence points vary with different "coincidence criteria".

In the following we consider few different pairs of orbits that can be used for cross-calibration based on existing and planned Ka and W-band radar systems.

### 2.1.1   Coincident overpasses for Ka-band conically scanning radars

At this band we assume that the GPM KaPR will be used as calibrator. The cross-track scanning radar is carried on a 65°
inclined orbit at 407 km altitude. Two types of orbits (one polar and one tropical, no specific information is currently available on the orbits of the constellation) are used to demonstrate the methodology for the cross-calibration of the radars of the Tomorrow.io constellation. The first is characterized by an altitude of 500 km and an inclination of 50°, and the other by a sun-synchronous orbit and an altitude of 500 km. The two combinations of the Tomorrow.io orbits with the GPM orbit define the first two orbital cross-over configurations. The orbital elements and the instrument specifics are reported in Tab. 2.
The calculation of the intersection is split in two steps: first the time intervals where the spacecrafts are close enough are computed (left panels in Fig. 2); then, in correspondence to the segments of orbits found in the first step, the positions of the antenna boresights at the ground is simulated with fine resolution so that the number of coincident footprints can be computed for any given coincidence criterion (right panels in Fig. 2). This is demonstrated in the upper panels of Fig. 2 for the





**Table 2.** Specifics of the Tomorrow.io and GPM satellites orbits and instruments. The radar that is used as a reference because it is properly calibrated is in the grey box. The Ka-band GPM radar is expected to be operational at the end of the decade.

| Radar | Tomorrow.io1 | Tomorrow.io2 | GPM KaPR |
|---|---|---|---|
| **Orbital elements** | | | |
| Eccentricity | 0.00125 | 0 | 0 |
| Semi-major axis [km] | 6878 | 6778 | 6785 |
| Inclination [deg] | 97.400 | 50 | 65 |
| RAAN [deg] | $-169.3870$ | 0 | 0 |
| Argument of periapsis [deg] | 90 | 0 | 0 |
| Mean anomaly [deg] | 90 | 0 | 0 |
| Mean LTAN [hour] | 6.000 | - | - |
| Epoch $t_0$ | 2019-01-01 06:00:00 | | |
| Reference Frame | J2000 | | |
| **Instrument specifics** | | | |
| RF output frequency | Ka band | | |
| Scanning type | Conical | | Cross-track |
| Swath width [km] | 400 | 400 | 245 |
| Off-nadir pointing angle [deg] | 38° | | 0°-17° |
| Rotating velocity [rpm] | 12 | | - |

intersections between Tomorrow.io2 and GPM. The number of Tomorrow.io monthly coincidence for the polar and inclined

Tomorrow.io and the GPM satellite are shown in the top and bottom left panels of Fig. 3, respectively. They have maxima of occurrences around the extreme latitudes associated to the GPM and the inclined Tomorrow.io orbits, respectively. For the polar orbiting Tomorrow.io some strongly longitude-dependent patterns appear due to the specific combination of orbits with the GPM satellite.

### 2.1.2   Coincident overpasses for W-band conically scanning radars

The NASA AOS mission envisages to launch two spacecrafts operating at 400 km altitude with a 50° orbit inclination, and at 450 km altitude on a sun-synchronous orbit, respectively. Both spacecrafts will carry a nadir pointing W-band atmospheric radar that can be used as reference calibrator. The Wivern mission plans to fly a satellite in a 500 km altitude and sun-synchronous circular orbit, carrying a conically scanning W band atmospheric radar (Illingworth et al., 2018). Detailed orbital parameters and instrument specifics are listed in Tab. 3. The two combinations of the Wivern with the two AOS orbits define the third and

the fourth orbital cross-over configurations. An example of orbit intersection is shown in the bottom panels of Fig. 2 whereas the number of Wivern monthly coincidence between AOS1 e AOS2 and the Wivern satellites are shown in the top and bottom right panels of Fig. 3, respectively. For the polar sun-synchronous AOS2 configuration intersections with Wivern are found

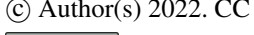



**Table 3.** Specifics of the AOS and Wivern satellites orbits and instruments. The radars that are used as a reference are inserted in the grey boxes.

| Radar | Wivern | AOS1 | AOS2 |
|---|---|---|---|
| **Orbital elements** | | | |
| Eccentricity | 0.00125 | 0 | 0 |
| Semi-major axis [km] | 6878 | 6778 | 6820 |
| Inclination [deg] | 97.400 | 50 | 97.213 |
| RAAN [deg] | $-169.3870$ | 0 | 122.922 |
| Argument of periapsis [deg] | 90 | 0 | 0 |
| Mean anomaly [deg] | 90 | 0 | 0 |
| Mean LTAN [hour] | 6.000 | - | 1.500 |
| Epoch $t_0$ | 2019-01-01 | | |
| Reference Frame | J2000 | | |
| **Instrument specifics** | | | |
| RF output frequency | W band | | |
| Scanning type | Conical | No scanning | |
| Swath width [km] | 800 | - | |
| Off-nadir pointing angle [deg] | 38° | 0° | |
| Rotating velocity [rpm] | 12 | - | |

only between 68 and 82 degrees latitude and peaking at the highest latitudes; viceversa, the intersections between the inclined AOS1 and Wivern are more likely to occur at the highest latitudes touched by AOS1 around 48 degrees latitude.

## 2.2 Calibrating targets (step2)

In this section the natural targets that can be used as a reference for cross-calibration are defined. The selection goes to ice clouds away from deep convection because such clouds are characterized by low attenuation both at Ka and W-band (Protat et al., 2019; Tridon et al., 2020) therefore their reflectivities do not change with different observation geometries (i.e. the measured reflectivities of an ice cloud observed at nadir and at slant incidence angles are almost identical). Different selection criteria are used at the two bands because of the different sensitivities of the reference radars.

### 2.2.1 Ka-band conically scanning radars

For Ka-band radars ice clouds with reflectivity exceeding 15 dBZ (a sensitivity that should be achieved by the Tomorrow.io radars), located at least 500 m above the freezing level and/or the surface and and with thicknesses exceeding one kilometre have been used. The Ka-band GPM radar the KaPR high sensitivity L2 products (zFactorMeasured, binZeroDeg, binClutterFreeBottom, https://gportal.jaxa.jp/gpr/assets/mng_upload/GPM/GPM_Product_List.pdf) have been used to character-





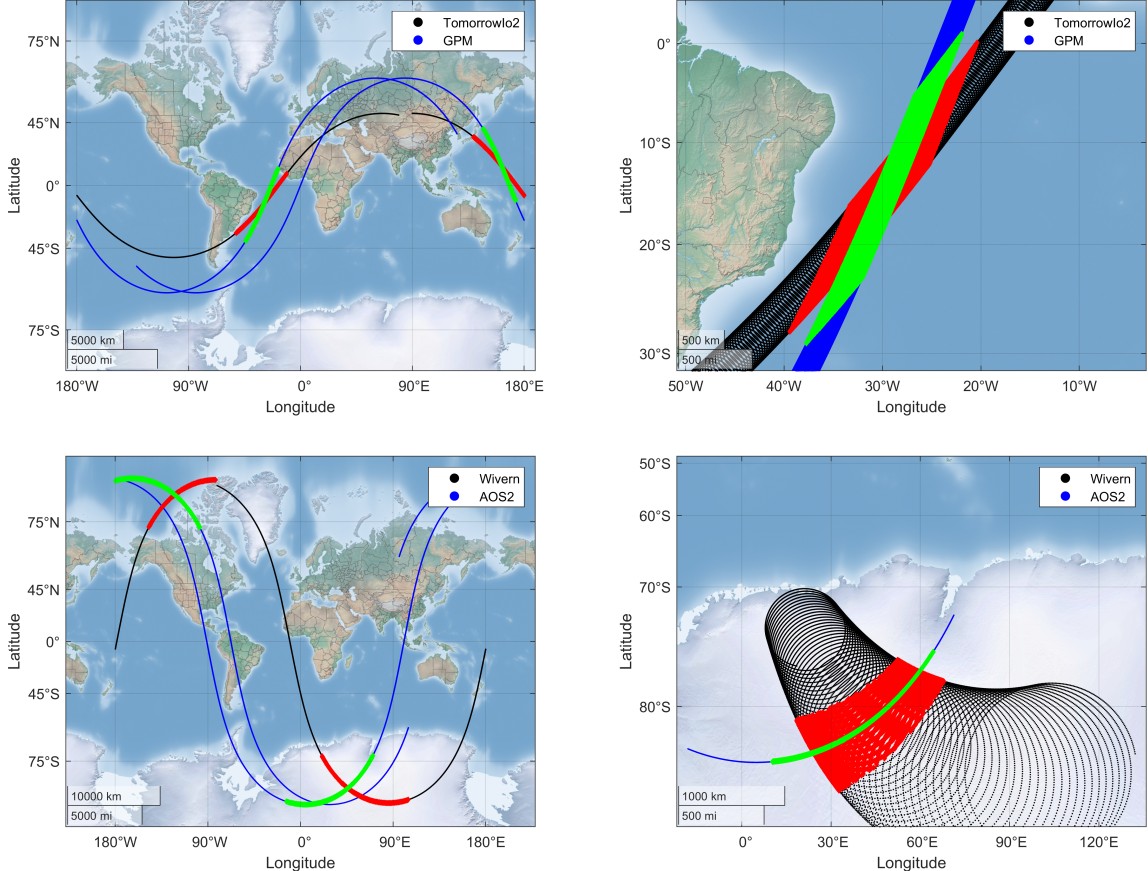

**Figure 2.** Intersection between the inclined Tomorrow.io and the Ka-DPR radar (top panels) and the AOS2 polar W-band radar and the conically scanning Wivern radar (bottom panels) according to criterion #7. The left panels represent the ground tracks of the two orbits whereas the right panels depict the details of the two radar footprints at the ground in the region where the ground tracks intersect.

ize where such clouds are located and how frequently they occur. An example of a mid-latitude frontal stratiform system observed by the GPM-Ka radar and of the corresponding ice calibrating clouds is depicted in Fig. 4a. The climatology of the mean number of the 500 m-thick radar bins of ice calibrating clouds is presented in the top panel of Fig. 5. These statistics are based on more than 4 years of GPM-Ka band data, from the satellite launch in 2014 to the 21st of May 2018 when the

scanning strategy was modified (see Liao and Meneghini, 2022) The figure shows patterns with maxima and minima in line with the climatology of ice water path derived from CloudSat and Calipso reported in Hong and Liu (2015, Fig.3). Thick ice clouds are frequently observed in regions of deep convection (e.g. the Inter Tropical Convergence Zone) and along storm tracks in the mid-latitudes. Note that in some locations in the Tropics, where thick ice clouds are ubiquitous, the mean number of





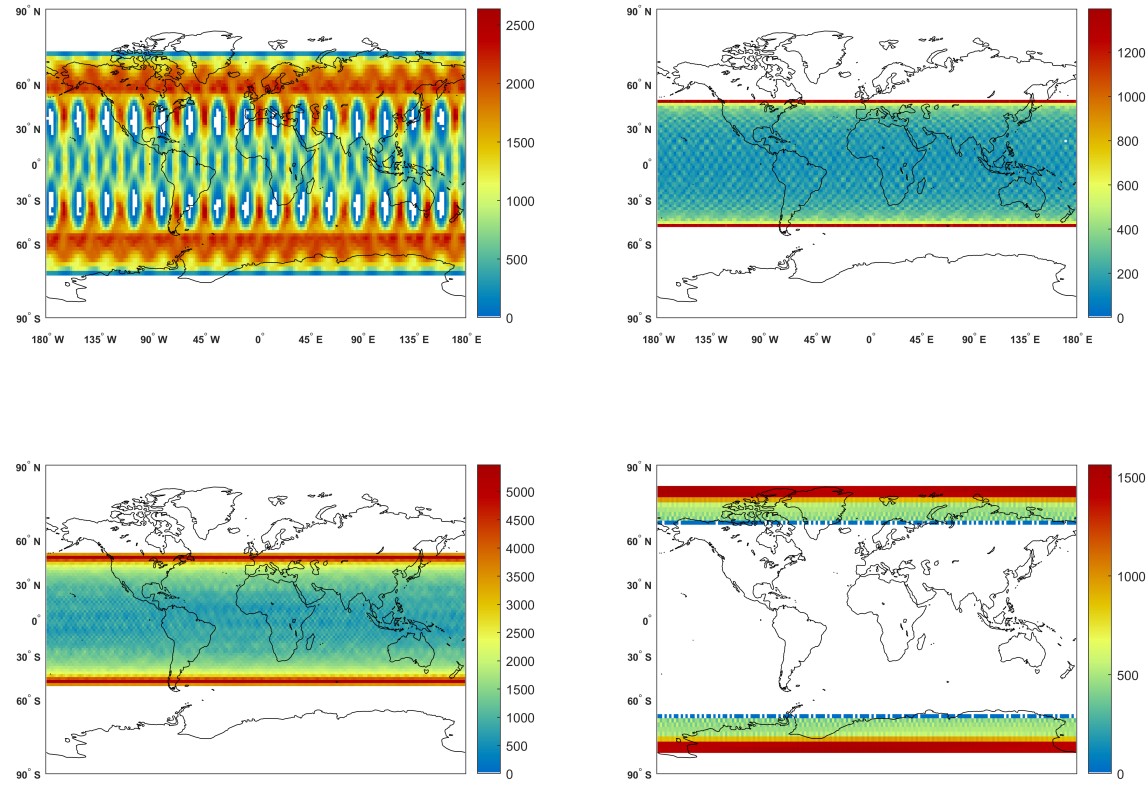

**Figure 3.** Number of monthly intersections between Tomorrow.io polar and inclined - GPM (top left, bottom left) and Wivern - AOS inclined and polar (top right, bottom right) according to criterion #9 (1000 km and 30 min). Each box has a resolution of $2° \times 2°$.

500 m-thick ice calibrating clouds is higher than one. This means that an intersection occurring in such areas will produce on

150 average more than one calibrating point.

Zonal plots of seasonal variation of mean number of 500 m-thick ice calibrating clouds are plotted in the left panel of Fig. 6 for the four seasons DJF (December-January-February), MAM (March-April-May), JJA (June-July-August), SON (September-October-Novemeber). In the tropics it is clear that the maximum moves with the intertropical convergence zone from south, during DJF, to the north of the Equator in JJA. These numbers represent multiplicative zonal factors that needs to be applied to

155 the number of satellite intersections to find the number of calibrating points.

The total number of weekly calibration points (obtained by multiplying the weekly number of intersection and the clima-tological number of 500 m-thick ice calibrating clouds) involving the two Tomorrow.io radars is reported in the four columns



**Table 4.** Number of weekly calibrating points. For each configuration the number of calibrating points for each of the radars involved in the cross-calibration are reported in the two corresponding columns. A calibration point correspond to a 5 and 1 km along-footprint track for the Ka and W-band, respectively.

| Criterion \ Configuration | Tomorrow.io1-GPM | | Tomorrow.io2-GPM | | Wivern-AOS1 | | Wivern-AOS2 | |
|---|---|---|---|---|---|---|---|---|
| | # calibrating points per week | | | | | | | |
| 1 | $2.09{\times}10^4$ | $4.11{\times}10^4$ | $5.66{\times}10^4$ | $1.06{\times}10^5$ | $3.99{\times}10^5$ | $3.29{\times}10^4$ | $1.40{\times}10^5$ | $1.09{\times}10^4$ |
| 2 | $3.09{\times}10^4$ | $5.53{\times}10^4$ | $8.51{\times}10^4$ | $1.45{\times}10^5$ | $8.07{\times}10^5$ | $4.00{\times}10^4$ | $3.01{\times}10^5$ | $1.28{\times}10^4$ |
| 3 | $6.13{\times}10^4$ | $1.09{\times}10^5$ | $1.80{\times}10^5$ | $2.96{\times}10^5$ | $2.08{\times}10^6$ | $6.15{\times}10^4$ | $8.00{\times}10^5$ | $1.97{\times}10^4$ |
| 4 | $1.16{\times}10^5$ | $2.02{\times}10^5$ | $3.67{\times}10^5$ | $5.51{\times}10^5$ | $4.36{\times}10^6$ | $1.00{\times}10^5$ | $1.80{\times}10^6$ | $4.14{\times}10^4$ |
| 5 | $2.41{\times}10^5$ | $4.21{\times}10^5$ | $8.85{\times}10^5$ | $1.14{\times}10^6$ | $9.60{\times}10^6$ | $1.92{\times}10^5$ | $5.12{\times}10^6$ | $9.94{\times}10^4$ |
| 6 | $3.87{\times}10^4$ | $7.53{\times}10^4$ | $1.11{\times}10^5$ | $2.07{\times}10^5$ | $7.91{\times}10^5$ | $6.34{\times}10^4$ | $2.80{\times}10^5$ | $2.12{\times}10^4$ |
| 7 | $5.73{\times}10^4$ | $1.02{\times}10^5$ | $1.67{\times}10^5$ | $2.81{\times}10^5$ | $1.59{\times}10^6$ | $7.63{\times}10^4$ | $5.97{\times}10^5$ | $2.48{\times}10^4$ |
| 8 | $1.13{\times}10^5$ | $1.97{\times}10^5$ | $3.47{\times}10^5$ | $5.62{\times}10^5$ | $4.01{\times}10^6$ | $1.15{\times}10^5$ | $1.55{\times}10^6$ | $3.79{\times}10^4$ |
| 9 | $2.09{\times}10^5$ | $3.61{\times}10^5$ | $6.85{\times}10^5$ | $1.02{\times}10^6$ | $8.16{\times}10^6$ | $1.83{\times}10^5$ | $3.40{\times}10^6$ | $7.71{\times}10^4$ |
| 10 | $4.15{\times}10^5$ | $7.18{\times}10^5$ | $1.56{\times}10^6$ | $1.99{\times}10^6$ | $1.69{\times}10^7$ | $3.31{\times}10^5$ | $9.28{\times}10^6$ | $1.80{\times}10^5$ |
| 11 | $5.65{\times}10^4$ | $1.10{\times}10^5$ | $1.64{\times}10^5$ | $3.01{\times}10^5$ | $1.14{\times}10^6$ | $9.03{\times}10^4$ | $4.21{\times}10^5$ | $3.15{\times}10^4$ |
| 12 | $8.34{\times}10^4$ | $1.48{\times}10^5$ | $2.44{\times}10^5$ | $4.08{\times}10^5$ | $2.28{\times}10^6$ | $1.09{\times}10^5$ | $8.96{\times}10^5$ | $3.68{\times}10^4$ |
| 13 | $1.63{\times}10^5$ | $2.87{\times}10^5$ | $5.03{\times}10^5$ | $8.14{\times}10^5$ | $5.73{\times}10^6$ | $1.63{\times}10^5$ | $2.31{\times}10^6$ | $5.66{\times}10^4$ |
| 14 | $3.01{\times}10^5$ | $5.21{\times}10^5$ | $9.80{\times}10^5$ | $2.46{\times}10^6$ | $1.15{\times}10^7$ | $2.55{\times}10^5$ | $4.93{\times}10^6$ | $1.12{\times}10^5$ |
| 15 | $5.77{\times}10^5$ | $9.96{\times}10^5$ | $1.83{\times}10^6$ | $2.35{\times}10^6$ | $2.33{\times}10^7$ | $4.46{\times}10^5$ | $1.28{\times}10^7$ | $2.49{\times}10^5$ |

(from the second to the fifth) in Tab. 4. Note how the number of calibrating points for GPM and for Tomorrow.io is similar (due to the similar sampling rate) and, as expected, is constantly increasing with $\Delta t$ and $\Delta r$.

### 2.2.2 W-band conically scanning radars

For W-band radars we have used ice clouds with reflectivity exceeding -20 dBZ (Wivern radar will certainly achieve such a sensitivity, Illingworth et al. (2018)) located in regions with temperature colder than 250 K, with a vertical extension of at least 750 m and located more than 2 km above clutter. The CloudSat CPR (with a much better sensitivity of around -30 dBZ) dataset can be exploited to understand where such clouds are located and how frequently they occur. Different Cloudsat products (see https://www.cloudsat.cira.colostate.edu/) are used to derive geo-located reflectivity profiles and to identify the surface clutter height (2B-GEOPROF), to determine the temperature profile (ECMWF-AUX) and to exclude deep convective cores (2B-CLDCLASS). An example of a tropical system observed by CloudSat and of the corresponding ice calibrating clouds is depicted in Fig. 4b.

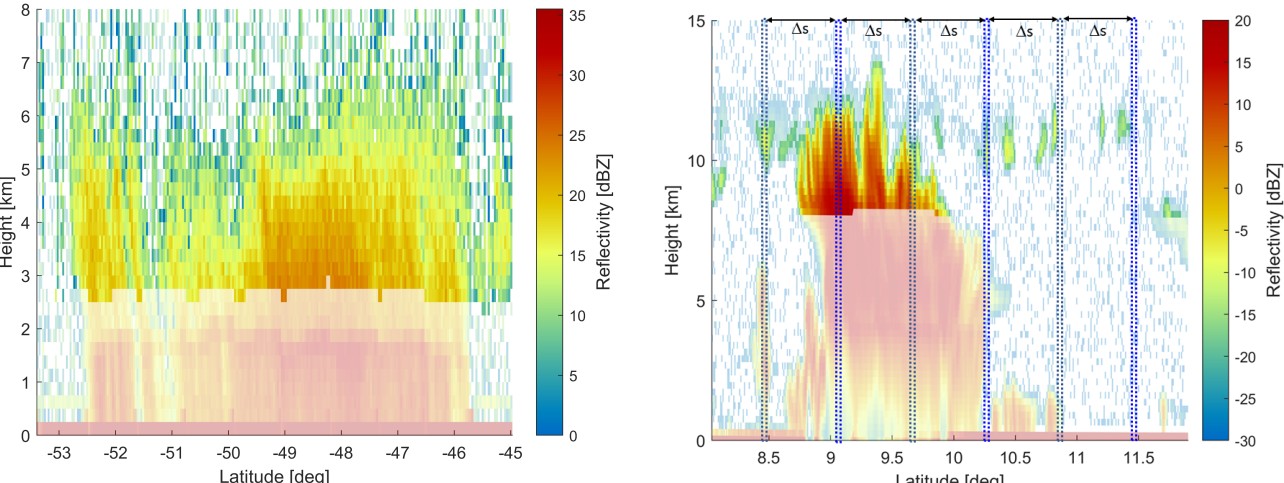

**Figure 4.** Example of GPM Ka-band (left) and CloudSat W-band (right) reflectivity profiles over a mid-latitude and a tropical system, respectively. The regions with bright colors represent the points that are used as natural target calibrating points. In the right panel the procedure adopted in step 3 is illustrated (see description in Sect. 2.3 for details). Each dashed box represent a 5 km along track slice. The reflectivities of the calibrating points are clustered together in correspondence to the black and blue boxes, which are separated by a given distance $\Delta s$ (for this specific example of the order of 500 km).

Four years of data from 2007 to 2010 have been analysed to compute the climatology of such clouds as observed by the CloudSat radar. The global distribution of the mean number of 500 m-thick ice W-band calibrating clouds is shown in Fig. 5. As expected, the patterns are very similar to those shown for Ka-band calibrating clouds. Also the mean number of ice layers are not very dissimilar because the better sensitivity at W-band is offset by the tighter constraint on the temperature and on the thickness of the layer. The zonal plots of the seasonal variation of the mean number of 500 m-thick ice calibrating clouds (right panel of Fig. 6) are similar to the results found at Ka-band as well. An alike seasonal movement with the intertropical convergence zone is also observed. Thicker ice clouds occur in northern and high latitudes during the warm season (JJA) and the Autumn (SON) with thinner clouds observed more frequently during the cold season (DJF) and the spring (MAM). In contrast, southern mid- and high latitudes have less variable ice cloud frequencies over the whole year.

The total numbers of calibration points involving the Wivern radar are reported in the four columns from the sixth to the ninth in Tab. 4. Thanks to its much higher footprint ground velocity ($\approx 800$ kms$^{-1}$) and its faster sampling rate, Wivern calibrating points are significantly more than those obtained for the AOS radars (with increasingly larger differences for larger $\Delta r$).

## 2.3 JS distances for biased and unbiased reflectivity PDFs (step 3)

The GPM and CloudSat datasets have been further exploited to determine what is the difference between the reflectivity PDFs when sampling calibrating clouds separated by a given distance. The procedure is illustrated in the right panel of Fig. 4: all ice





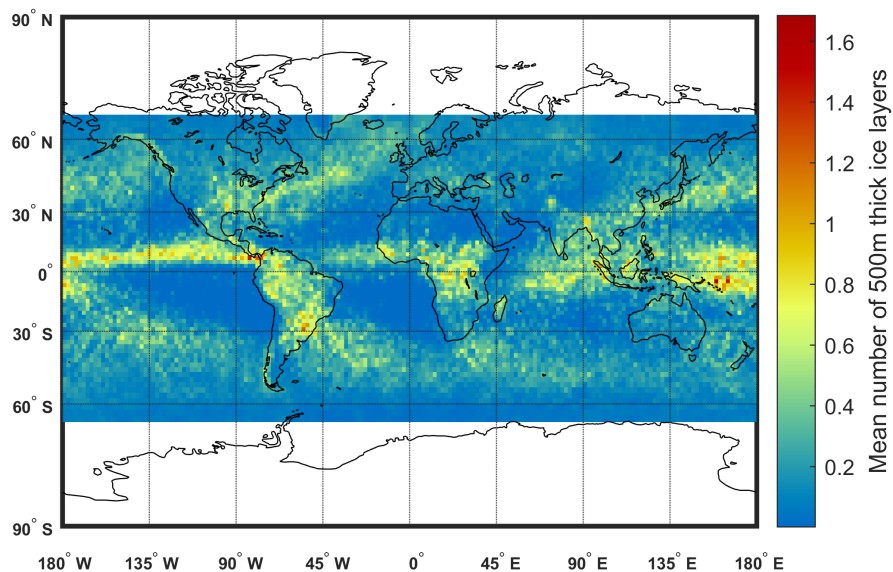

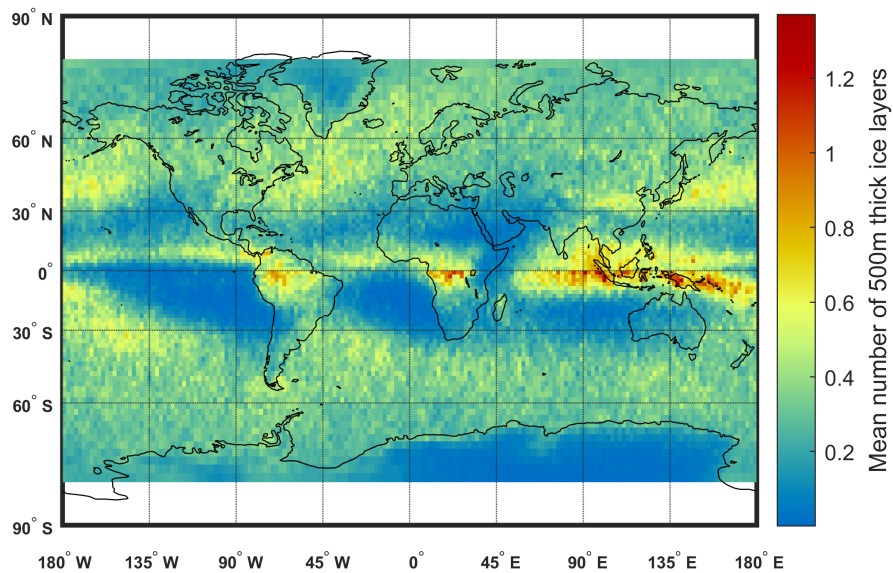

**Figure 5.** Global distribution of the mean number of 500 m-thick layers of Ka-band (top) and of 500 m-thick layers of W-band (bottom) ice calibrating clouds as derived by 4 years of GPM and CloudSat data, respectively. The resolution is $2° \times 2°$.



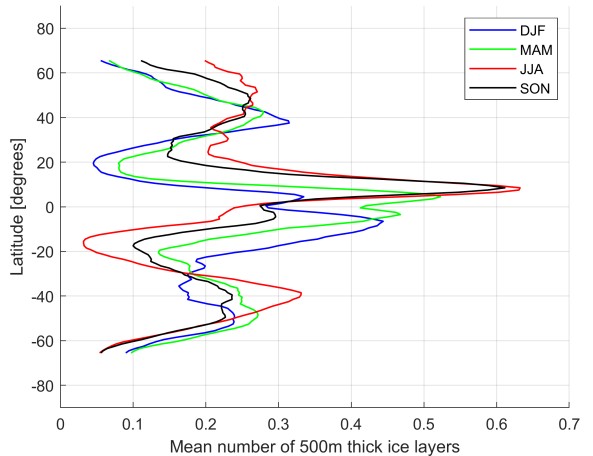
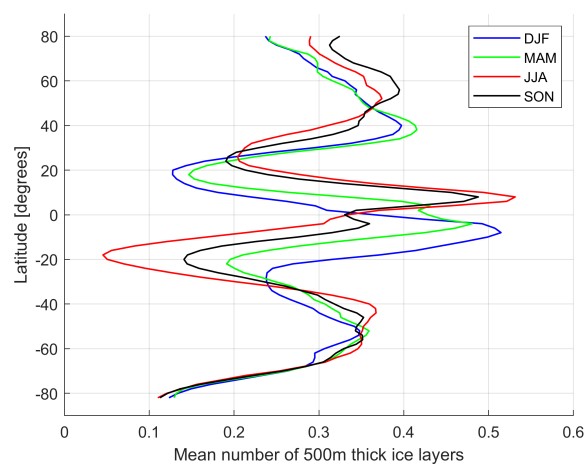

**Figure 6.** Zonal variability of the mean number of 500 m thick ice layers for Ka-band and W-band ice calibrating clouds, respectively. Four different seasons are plotted, as indicated in the legend. The same datasets adopted to produce Fig. 5 have been used.

cloud radar measurements within the black and blue 5 km wide vertical slices are accumulated separately. The reflectivities

within the blue and the black slices representing samples separated by $\Delta s$ are clustered to produce two independent reflectivity PDFs. The PDFS are accumulated until a given sample size is reached (which will correspond to given measurement period). Then the JS distance between the two PDFs is computed (see Eq. 2). The procedure is repeated by constructing other independent pairs of Z-PDFs. Each Z-PDF pair will be characterized by a different JS distance.

Examples of a pair of Z-PDFs are shown in Fig. 7 for both Ka and W-band reflectivities when including 50,000 calibration

points (red and blue lines). The median global climatology PDF is plotted as a dashed black line whereas the 5th and 95th percentile of the PDF is shown by the grey shading. Note how the Z-PDF monotonically decreases both at Ka and W-band with increasing reflectivity values. While the spread between the high and low percentiles decreases with increasing Z, the relative noisiness of the PDFs increases when going towards Z-values with low probability of occurrence. These values will not contribute much in the JS distances because of the multiplicative factor $P(x)$ in Eq. 3. For the W-band PDF the effect

of a positive bias of 1 and 2 dB is also illustrated by the green and the yellow curve, respectively. The two curves depart significantly from the median behaviour and they dwell outside the shaded regions for a number of points significantly larger when increasing the bias from 1 to 2 dB. For instance, for the example shown in the right panel, the JS distance between the median PDF (black dashed line) and the blue, red, green and yellow curves is equal to 0.0338, 0.0334, 0.0476, 0.0646, respectively.

In order to understand what are the JS distances expected when comparing PDFs of reflectivities collected by two radars from clouds at a given separation distances, the previously described procedure is repeated for different sample sizes and for an ensemble of pairs large enough to characterize the distribution function of the JS distances for any given sample size and separation, $\Delta s$. The median value (black line) of the JS distance for the ensemble of pairs of Ka and W-band Z-PDFs is shown





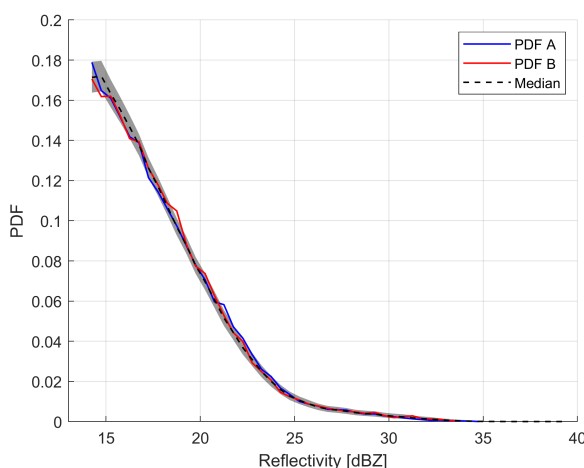
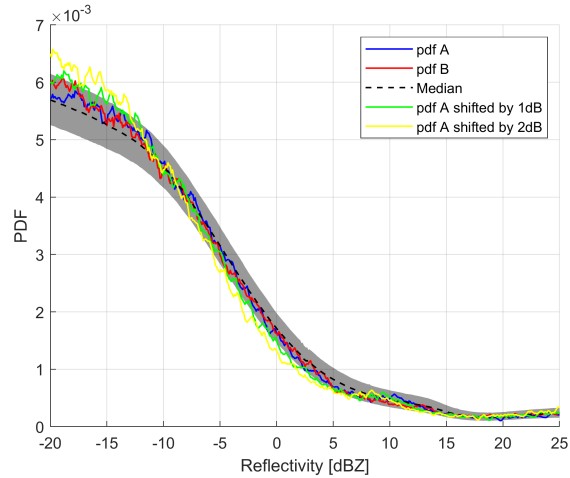

**Figure 7.** Envelope (5-95$^{th}$ percentile) of Z-PDFs for the Ka-band ice calibrating clouds (left panel) and for the W-band ice calibrating clouds (right panel). The black line represents the median Z-PDF whereas the blue and red curve represent two PDFs randomly selected. Both PDFs have been built with about 50,000 calibration points (Tab. 4 can be used to compute the time needed to collect such sample according to the different configurations and coincidence criteria). For the W-band PDF the effect of a positive bias of 1 and 2 dB is also shown.

in Fig. 8 as a function of the number of calibrating points for a separation distance of 500 and 100 km, respectively. The
dashed black lines indicate the $5^{th}$ and $95^{th}$ percentiles of the ensemble. As expected, the median values of the JS distances
(continuous black line) decrease with increasing number of calibrating points (i.e. the PDFs become less and less noisy). The
envelop between the dashed lines well represents the range of values of the JS distances between Z-PDFs for ice calibrating
clouds separated by 500 km at Ka-band (left) and by 100 km at W-band (right) for different numbers of calibrating points.

The same exercise is repeated by shifting one of the two reflectivity Z-PDFs of the pair by a calibration reflectivity bias
(e.g. 0, $\pm 0.5$ dB, $\pm 1$ dB, . . . ) to assess what is the impact of a miscalibration onto the JS distances. All JS distances jump
up with a shift that increases with the bias magnitude, as demonstrated in Fig. 8 where positive biases of +0.5, +1 and +2 dB
are considered. Similar results are obtained when negative biases are applied. From the plot it is clear that, for each value of
the reflectivity bias, there is a threshold of calibration points above which the range of values in the corresponding coloured
envelope is clearly distinct from the unbiased range delimited by the black dashed lines. For the three biases shown in the right
panel of Fig. 8 these values are indicated by the numbers $N_{0.5}$, $N_1$ and $N_2$ with $N_{0.5} > N_1 > N_2$. For instance, with 100,000
calibration point defined by a separation distance of less than 100 km, a bias of 2 dB produces JS distances above 0.038 ($5^{th}$
percentile value) which are incompatible with the range of values expected from the natural variability between 0.019 and 0.028
(respectively $5^{th}$ and $95^{th}$ percentiles, dashed black lines). On the other hand a 1 dB bias is expected to produce JS distances
between 0.024 and 0.037 and therefore cannot be unequivocally identified. Collecting a sample larger than $N_1 = 187,210$
would guarantee to be able to identify a bias of 1 dB. A much larger value ($N_{0.5} = 667,700$) would be required to discern a
miscalibration of 0.5 dB. Also the red envelope remains very close to the envelope identified by the dashed black lines with





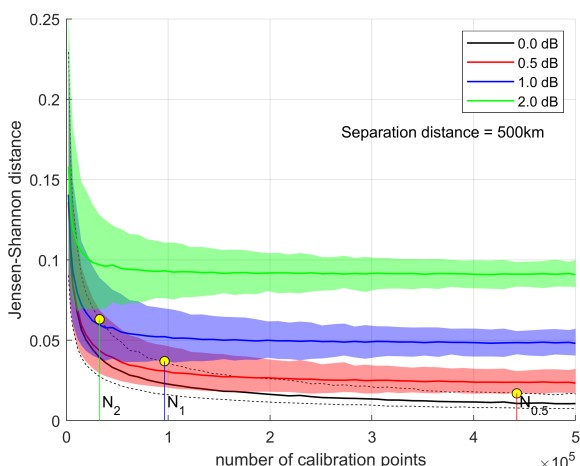
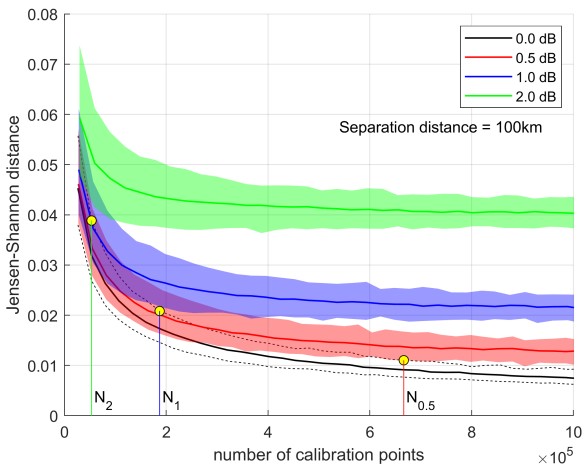

**Figure 8.** Characterization of the JS distance for Ka (left) and W-band (right) Z-PDFs for ice calibrating targets separated by $\Delta s = 500\,\mathrm{km}$ as a function of the number of calibrating points. The median distance is indicated by the black line whereas the dashed black lines indicate the $5^{th}$ and $95^{th}$ percentiles. The coloured lines with the shaded envelopes correspond to the median and the same percentiles when shifting one of the two Z-PDF by different reflectivity biases as indicated in the legend. Here positive biases are shown but similar results are obtained when negative biases are applied.

increasing numbers of calibrating points which makes the calibration method more difficult to be applied at such high accuracy levels.

Therefore the plots in Fig. 8 can be used to understand what reflectivity biases are discernible when collecting a certain number of reflectivities corresponding to calibration points located within a given distance. Generally speaking the calibration for the Ka-band radars is performing better than the one for W-band radar (i.e. a smaller number of calibrating points is needed to achieve a given calibration accuracy). This is due to the sharper sloping of the Z-PDF of the Ka-band calibrating targets compared to that of the Z-PDF of the W-band calibrating targets [a factor of 10 decrease in 10 (30) dB reflectivity range for Ka-(W-)band]. With our method, a constant Z-PDF would be useless whereas a square wave-shape Z-PDF would perform

optimally.

     Results for the JS distances as a function of the number of calibrating points when changing $\Delta s$ are shown in Fig. 9. For the same number of calibrating points, smaller $\Delta s$ are characterised by a smaller J-S distance, as expected by an increased correlation of the reflectivities, aggregated at shorter separation distances, between the two sampled Z-PDFs. Note that the variability between the median curves (continuous lines) corresponding to different separation distances tend to be of the

same magnitude as the variability between the $5^{th}$ and $95^{th}$ percentiles of each single PDF (dashed envelopes) with the distances between the median curves becoming smaller and smaller at large separation distances. This feature suggests that, in a calibration procedure, it is more effective to use large values of $\Delta s$ because this choice will enable to collect the same number of calibrating points in a quicker time.



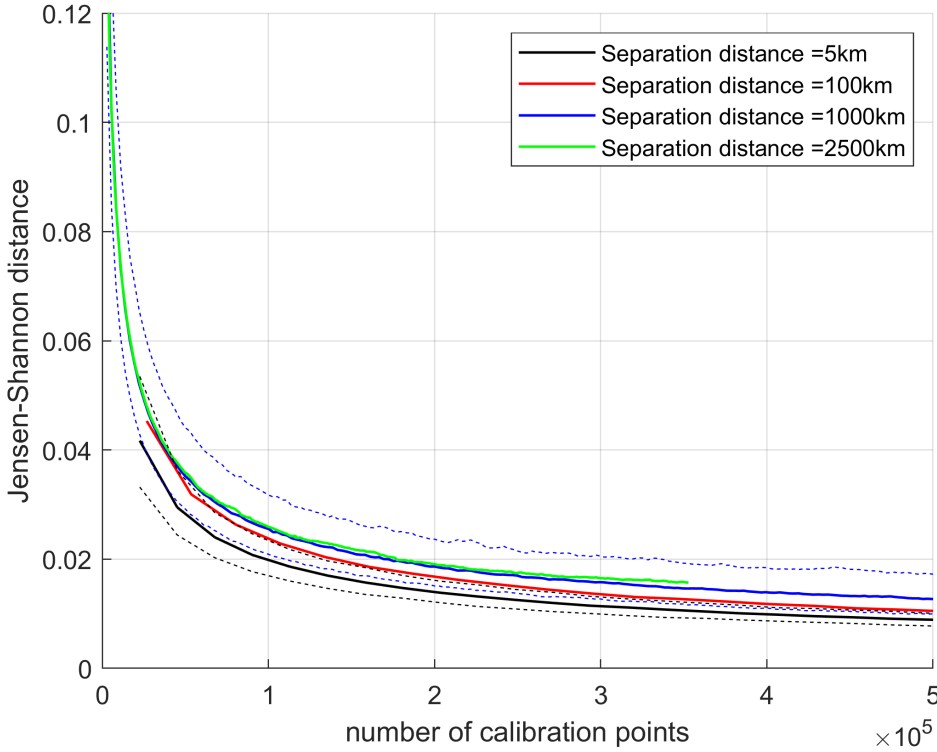

**Figure 9.** Impact of the separation distance on the J-S distances as a function of the number of points for the PDFs of W-band calibrating targets. The four continuous coloured lines correspond to the median of the J-S distances for different separation distances, as indicated in the legend. The black and the blue dashed lines correspond to the $5^{th}$ and $95^{th}$ percentiles for the 5 and the 1000 km separation distances.

## 3    Results of cross-calibration performance

The results of step 3 and step 4 can be combined to assess what accuracy can be achieved in the cross-calibration in a given time period and which is the optimal coincidence criterion to be used among those listed in Tab. 1. For each criterion and for each reflectivity bias the number of calibration points above which the envelopes of the unbiased and biased JS distances become distinct is computed. These numbers are converted in number of days required to collect such calibration points based on the figures tabulated in Tab. 3. In a conservative approach, for each radar-pair configuration, the lowest number of the two

columns is used. This is particularly penalizing for the Wivern system but it is a consequence of the fact that the noisiness of the JS distances will be driven by the reduced sampling capability of the AOS radars.

Results show that criteria based on large $\Delta s$ are generally preferable, i.e. the increased number of coincidences when adopting criteria with large $\Delta s$ tend to overcome the reduced correlation of cloud Z-PDFs. Thanks to the large swath of GPM and Tomorrow.io radars, for Tomorrow.io radars cross-calibrations with GPM Ka-band radar within 1 dB are realistically

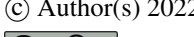



**Table 5.** Mean number of days required to achieve a given calibration accuracy for the different criteria listed in Tab. 1 and for the four configurations analysed in this study. A symbol – means that the given level of accuracy in the calibration cannot be achieved. Results with positive and negative mis-calibration are very similar. Here for concision only positive miscalibrations are considered.

| Configuration | Tomorrow.io1-GPM | | | Tomorrow.io2-GPM | | | Wivern-AOS1 | | | Wivern-AOS2 | | |
|---|---|---|---|---|---|---|---|---|---|---|---|---|
| | Mean number of days required to achieve a calibration accuracy better than | | | | | | | | | | | |
| Criterion # \ Miscalibration | 0.5 | 1.0 | 2.0 | 0.5 | 1.0 | 2.0 | 0.5 | 1.0 | 2.0 | 0.5 | 1.0 | 2.0 |
| 1 | 121.0 | 27.2 | 9.8 | 44.7 | 10.0 | 3.6 | 142 | 39.9 | 1.14 | 426 | 120 | 34.2 |
| 2 | 85.7 | 20.0 | 6.53 | 31.1 | 7.3 | 2.4 | 124 | 35.7 | 9.5 | 385 | 111 | 29.7 |
| 3 | 50.5 | 11.0 | 3.73 | 17.2 | 3.7 | 1.3 | 140 | 31.8 | 7.5 | 436 | 99.4 | 23.4 |
| 4 | 29.9 | 6.1 | 1.7 | 9.5 | 1.9 | 0.5 | – | 21.8 | 6.2 | – | 52.7 | 14.9 |
| 5 | 17.8 | 3.4 | 0.8 | 4.6 | 0.9 | 0.2 | – | 12.4 | 3.0 | – | 23.9 | 5.7 |
| 6 | 65.3 | 14.7 | 5.3 | 22.7 | 5.1 | 1.8 | 73.6 | 20.7 | 5.9 | 220 | 61.8 | 17.7 |
| 7 | 46.2 | 10.8 | 3.5 | 15.9 | 3.7 | 1.2 | 64.8 | 18.7 | 5.0 | 200 | 57.6 | 15.4 |
| 8 | 27.4 | 6.0 | 2.0 | 8.9 | 1.9 | 0.7 | 74.4 | 17.0 | 4.0 | 227 | 51.7 | 12.2 |
| 9 | 16.7 | 3.4 | 1.0 | 5.1 | 1.0 | 0.3 | – | 11.9 | 3.4 | – | 28.3 | 8.1 |
| 10 | 10.3 | 2.0 | 0.5 | 2.8 | 0.5 | 0.13 | – | 7.2 | 1.7 | – | 13.2 | 3.2 |
| 11 | 44.8 | 10.1 | 3.6 | 15.4 | 3.5 | 1.2 | 51.7 | 14.5 | 4.1 | 148 | 41.5 | 11.9 |
| 12 | 31.7 | 7.4 | 2.4 | 10.8 | 2.5 | 0.8 | 45.6 | 13.2 | 3.5 | 135 | 38.8 | 10.4 |
| 13 | 19.0 | 4.1 | 1.4 | 6.2 | 1.3 | 0.5 | 52.6 | 12.0 | 2.8 | 152 | 34.6 | 8.2 |
| 14 | 11.6 | 2.4 | 0.7 | 3.6 | 0.7 | 0.2 | – | 8.5 | 2.4 | – | 19.6 | 5.6 |
| 15 | 7.4 | 1.4 | 0.4 | 2.4 | 0.4 | 0.1 | – | 5.3 | 1.3 | – | 9.5 | 2.3 |

achieved on average within few days (less than two when considering criterion#10 and #15), a very promising result for the Tomorrow.io constellation. Even a cross-calibration within 0.5 dB seems feasible within a week for the polar Tomorrow.io and few days for the inclined Tomorrow.io. In general cross-calibration with GPM can be performed on shorter timescales for the inclined Tomorrow.io configuration.

For Wivern, cross-calibrations with an accuracy better than 1 dB is feasible on time scales of the order of less than 7 days for AOS1 and of less than 10 days for AOS2 (both achieved with criterion #15). A calibration within 2 dB on the other hand can be achieved much more quickly (within two and three days for AOS1 and AOS2, respectively).

It is important to note that these are mean results based on annual climatology of clouds and of orbital intersections; in specific conditions, with lack of orbital intersections and absence of ice clouds, the time needed for achieving these results can be longer. However, for the criterion #15 that achieves the best results, there is small variability on a weekly basis, as demonstrated by the whisker plot in Fig. 10. The combination between the polar Tomorrow.io and GPM shows large variability from week to week with a cycle that exceeds one year. The largest variability is encountered with the inclined Tomorrow.io



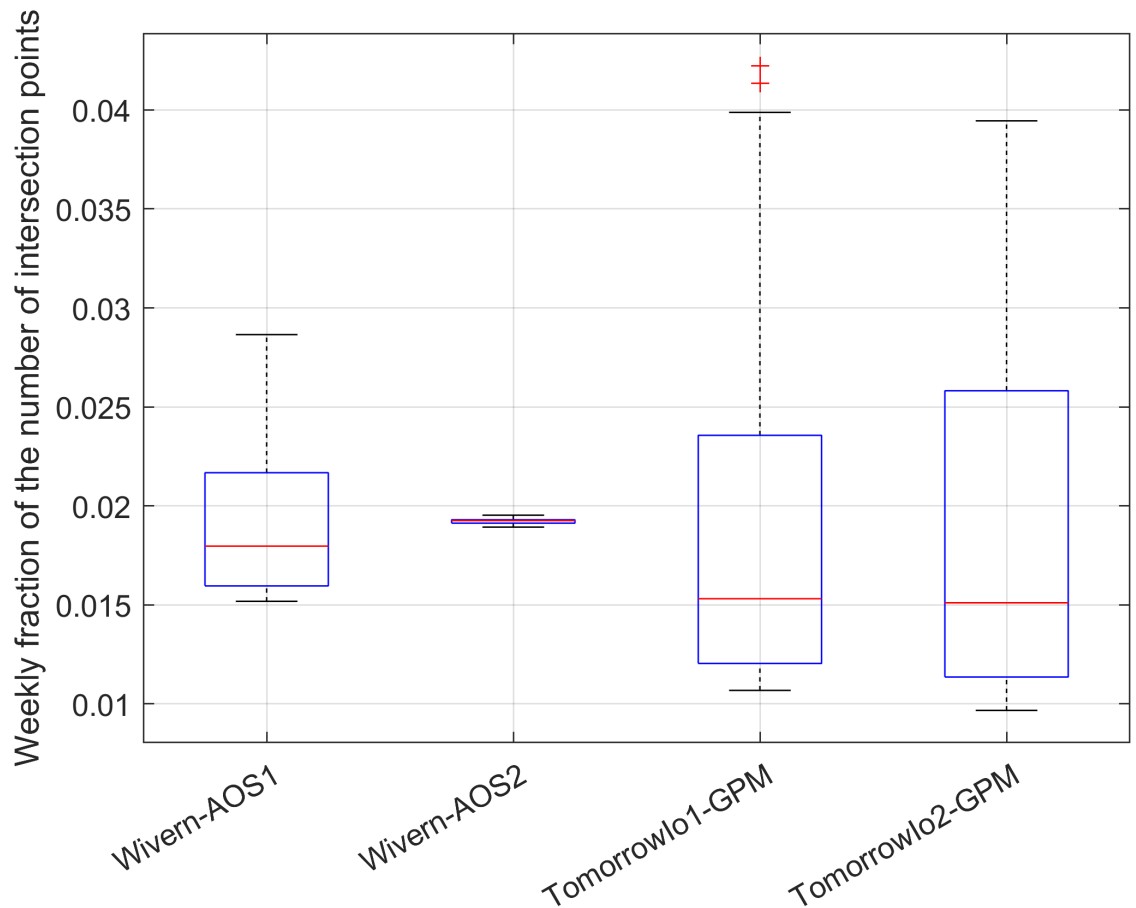

**Figure 10.** Whisker plot with the weekly fraction of the year calibration points for the 52 weeks of the year for the four satellite configurations considered in this study and for criterion #15. The annual number of calibration points are about 75, 63, 602, 743 millions for AOS1, AOS2, Tomorrow.io1 and Tomorrow.io2, respectively.

orbit. In the worst case scenario, for such configuration, some weeks produce less than 1% of the annual number of calibration points.

Since the best results are obtained with the largest temporal and spatial separation distance it is sensible to ask whether
the curves of the global climatology of ice clouds Z-PDFs could be used as absolute calibration curves, without the need of satellites' cross-over. In order to investigate this possibility the following steps are followed:

- a climatological Z-PDF ($PDF_1$) covering 4 years of data is produced;

- for each day in the dataset a separate Z-PDF is generated ($PSD_j$) and JS distances between $PDF_1$ and $PDF_j$ are computed at different aggregation intervals (weekly and monthly);





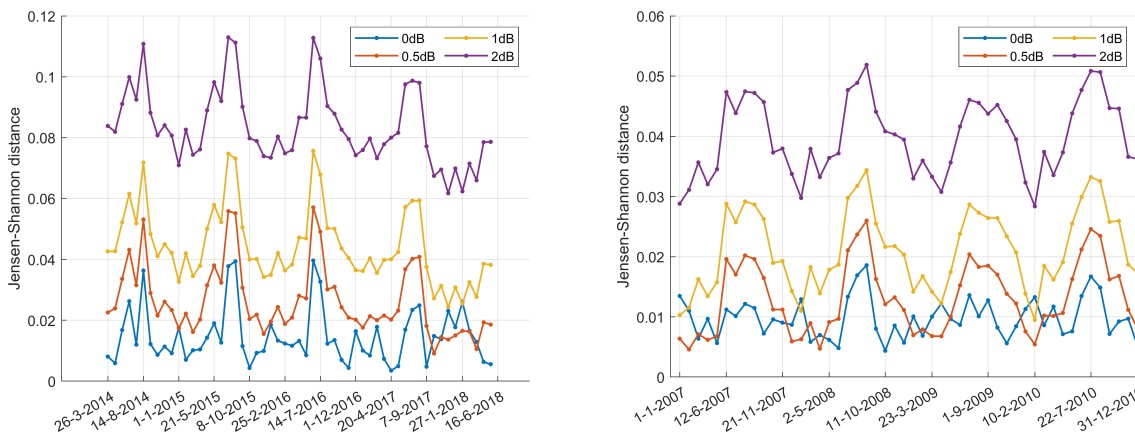

**Figure 11.** Time evolution of the JS distances of monthly cumulated Z-PDFs to four years of Ka (left) and W-band (right) climatology.

– $PDF_j$ is shifted by $\pm 0.5$, $\pm 1$ $\pm 2$ dB and the corresponding JS distances between $PDF_1$ and the six shifted PDFs are computed;

    – time series of the seven different JS distances are produced and compared.

Results accumulated at the monthly scale depicted in Fig. 11 for the Ka (left) and W-band (right panel) show that a climatological calibration seems feasible within 1 dB at such temporal scale (2 dB are attainable at the weekly scale, not shown).

The presence of an annual cycle is also evident both at Ka and at W-band. For the GPM Ka-band radar an anomalous behaviour appears after July 2017 with the 0.5 dB curve intersecting and becoming smaller in magnitude than the 0 dB curve. This climatological approach could therefore represent a way to monitor long-term calibration issues for a single radar system as well.

## 4   Summary and conclusions

This study presents a methodology for calibrating a spaceborne conically scanning radar by using cross-calibration with reference spaceborne radars working on the same band and orbiting around Earth at the same time. Example of such systems are the Ka-band GPM radar or the W-band radars planned for the ESA-JAXA EarthCARE or for the NASA AOS missions. Ice clouds at cold temperatures (not prone to appreciable attenuation) are used as natural targets that allow to cross-calibrate the systems.

Radar antenna boresight positions have been propagated based on satellite orbits; then, the ground-track intersections have been calculated for different intersection criteria, as defined by cross-overs within a certain time and a given distance. Then the climatology of the calibrating clouds has been studied using the W-band CloudSat and Ka-band GPM reflectivity dataset in order to derive the global distribution of the frequency of the ice calibrating clouds (climatology presented in Fig. 5). The



number and the spatial distribution of calibration points is finally obtained by multiplying the ground-track intersections and
the climatology of clouds.

The similarity between the reflectivity distribution functions of the calibrating ice clouds at different separation distances has
been studied using CloudSat and GPM dataset in order to find the optimal distance criterion which optimizes the calibration
accuracy and minimizes the time needed to achieve such an accuracy. This requires a trade-off between having a sufficiently
large number of observations to reach statistical significance (obtained by relaxing the coincidence criterion) and a reasonable
invariance of the cloud reflectivity statistical properties (achieved by tightening the coincidence criterion).

Findings of this work demonstrate that it will be possible to cross-calibrate within 1 dB a Ka-band (W-band) conically scan-
ning radar like that envisaged for the Tomorrow.io constellation (Wivern mission) every few days (a week). Such uncertainties
are generally meeting the mission requirements and the standards currently achieved with absolute calibration accuracy. The
better performances achieved for the Tomorrow.io is the result of the higher number of intersections (thanks to the combined
scanning pattern of the Tomorrow.io radars and GPM) and to a shape of the Z-PDF better suited to perform cross-calibration).

In principle the global climatology Z-PDF (black continuous lines in Fig. 7) could be used as an absolute reference and the
JS distance could be computed with respect to such PDF. This would completely remove the issue of having intersections but
it is not sure how the natural variability introduced by regional, diurnal and seasonal cycles could impact in the uncertainties
of the Z-PDF itself. More research needs to be done in this area, mainly hampered by the lack of global observations of the ice
cloud diurnal cycle.

Calibration of radar reflectivities are paramount for producing unbiased hydrometeor mass contents and mass fluxes, which
represent flagships products in most of the cloud and precipitation radar missions. The approach described in this work is
applicable to estimate the cross-calibration accuracy for any orbital cross-over and will be applicable already for the calibration
of the Ka-band Tomorrow.io and the INCUS train (Stephens et al., 2020) radars, expected to be launched starting from the end
of 2023 and in 2027, respectively.

*Data availability.* This research is based on CloudSat and GPM data that are publicly available at http://
www.cloudsat.cira.colostate.edu/data-products (Cloudsat data processing center, 2018) and at https://gpm.nasa.gov/data/
sources, respectively.

*Author contributions.* AB wrote most of the text and has defined the project. FES has implemented the orbit intersection modelling; FES
and KM have performed all the analysis for the W and Ka band climatology, respectively and they have thoroughly reviewed the text. AI
contributed to the discussion and the review of the paper.

*Competing interests.* The authors declare that they have no conflict of interest.



*Acknowledgements.* This work was supported in part by the European Space Agency under the activity WInd VElocity Radar Nephoscope (WIVERN) Phase 0 Science and Requirements Consolidation Study, ESA Contract Number 4000136466/21/NL/LF. AB's work was funded
by Compagnia di San Paolo. This research used the Mafalda cluster at Politecnico di Torino.



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
