# Peer review of "In orbit cross-calibration of millimeter conically scanning spaceborne radars"

_Atmospheric Measurement Techniques, 2022_

## Author Comment (AC1)

**REPLY to REVIEWER 1**

We thank the reviewer for their review and their detailed comments.
Below you will find the reviewer's comments in bold and our replies.

**My main concern is on the criteria used to find coincidences. It seems that the Authors considered surface footprint coincidences only (with a given imposed uncertainty in the definition of coincident foot prints as in their Tab. 1) between a reference nadir- (or quasi-nadir) pointing radar platform and a conically scanning one. However, in the studied configurations you could have coincidences not only in the ground footprints (i.e. the lon, lat surface level) but also aloft (lon, lat, altitude) whenever two radar ray paths intersects each other. I think that last case is more relevant when considering natural ice clouds for calibration. It would be nice if the Authors could better elaborate this point in the main text. In addition, as I can understand, two parameters are important in the definition of a calibrating natural target: $Z_{min}$ and $h_{min}$ being the former the sensitivity of the radar system whereas the latter the minimum altitude wrt. to the surface considered to identify an ice clouds. Maybe these two parameters could be added in the scheme of Figure 1.**
We consider this effect negligible. In fact the difference in the distance of the two intersections, one measured at a certain altitude from the ground and the other measured at the sea level, is negligible compared to the high distance criteria we introduced in the intersection criteria. In fact, if we consider the maximum altitude of an anvil cloud ($\sim 20$ km), the distance between the radar path intersection at that altitude and a footprint intersection at sea level would be 20 km, very small if compared to the 1000 and 2000 km distance constraints.

**In Section 2.2 you state that "the measured reflectivity of an ice cloud observed at nadir and at slant incidence angles are almost identical". In general, I agree with you but I am thinking that the slanted geometry maybe could be more prone.**
The comment seems to be incomplete. In general the reflectivity may be slightly different in presence of non-sperical particles which are preferentially oriented. However we tend to use reflectivity value that are large, i.e. which tend to correspond to randomly oriented aggregates, which tend to have backscattering cross sections independent from incidence angle. Attenuation can introduce a difference because of the difference between slant paths and nadir path but this could be minimised trying to avoid deep convection.

**With reference of results argued in figure 8, what happens if you also a random noise to the "biased measurements" of Z. Maybe this could help to check the calibration performances when assuming a different error structure in the system being calibrated.**
In Fig.8 the random noise is implicitly included in the figure because we are considering clouds sampled at different distances (500 and 1000 km in the left and right panel, respectively). So we are sampling different clouds.

**Do you have some evidence of scan strategy of Tomorrow-io radar, i.e. reference or personal communication? It will be nice to add them in the manuscript.** Unfortunately we have only some informal personal communications on this.

**L 35. The general statement: near nadir looking "normalised backscattering cross section is insensitive to changes of the wind speed and the wind direction" seems to disagree with common radar altimeters applications (es. AltiKa). AltiKa is a nadir looking altimeter in Ka band and it is sensitive to wind speed as testified by its products (https://space.oscar.wmo.int/instruments/view/altika). Maybe, in this case, the range resolution (hundred meters) of meteo Ka and W-band with respect to radar altimeters (order of cm)**

can play a role in the ocean wind insensitivity. **When you say that the W-band nadir-looking calibration procedures are impractical for conically scan systems, maybe you can mention that at slanted (>40°) off zenith angles, the sea surface is specular, isnt'it?**

That statement is referred to the condition where the incident angle is 10°. Therefore is not applicable to altimeter which are nadir looking.

**Section 2. Step1. Maybe the term "coincident footprints" is misleading because it recalls surface footprint. If I understand well you are interested to any "range path intersection" between vertically and slanted pointing radar system.**

We will use the term quasi-coincident overpasses to avoid confusion.

**L 85 there is a repetition in the word: "calibration".**

Corrected.

**Page 8, Figure 4a is not found. Please specify Figure 4, left panel, or add labels on figure 4.**

Corrected.

**Section 2.3. At the beginning of this section it is not clear to the reader why you should sample two PDFs at a given separation distance? I think that this is made to simulate differences in the sampling position when actual conical scan measurements will be available compared to the nadir-based ones. Please explain.**

By taking in consideration one of the intersection criteria, all points sampled by the two radars that are closer than the distance defined by the selected intersection criterion are intersection points. With this procedure we want to observe what is the similarity of clouds sampled at that specific distance defined by the selected intersection criterion, since we know that those clouds will not be identical. This procedure is needed to establish, with a statistical analysis, what is the number of intersection points required in order to detect a certain miscalibration in the radar with the selected intersection criterion. In fact, the looser the criterion is the larger the number of intersection points; however the clouds at that separation distance are less correlated and more points are needed to detect that certain miscalibration. We will include this explanation in the revised paper.

**Not clear of the PDFs A and B are those extracted by vertical slices 500 km apart.**

Yes this is the case. We will update the caption.

**Not clear, from the figure's legend, compared with main text and inner figure text, if the two panels differ from the frequency band considered only or they also differ by the separation distance considered in the selection of Z profiles.**

They also differ by separation distance. We updated the caption.

---

## Author Comment (AC2)

We thank the reviewer for their review and their detailed comments.
Below you will find the reviewer's comments in bold and our replies.

**Section 2: I'm not sure I agree with the methodology used to calculate the expected number of calibration points per unit time presented in the manuscript, and will outline my concerns here, though I welcome input from the authors clarifying if I have missed a step or am misunderstanding their approach. The first point I would like to clarify is that the quantity referred to as "monthly intersections" throughout the paper is in fact more aptly described as the monthly-average number of instrument footprints satisfying the coincidence criteria gridded at 2 x 2 degree resolution. That is to say, for a single intersection of the orbit tracks of the two spacecraft (assuming nearly temporal coincidence), there are many points that will satisfy the spatiotemporal criteria and be logged as many "intersections" in a single 2 x 2 deg grid cell. If this is true, the problem for me arises when this "weekly/monthly intersections" variable is multiplied by the "mean number of ice layers" variable to get the number of weekly/monthly calibrating points. Specifically, I interpret the distributions in figure 5 as the mean number of ice cloud range bins per 2 x 2 degree grid cell, and not the mean number per horizontal radar footprint gridded at 2 x 2 degrees. Thus, to properly calculate the number of coincident ice cloud range bin detections within a given 2 x 2 degree grid cell, I would calculate it as: (mean number of ice cloud bins per 2 x 2 deg)\*(number of footprints within the 2 x 2 deg grid cell satisfying the coincidence criteria per unit time)/(number of footprints per 2 x 2 deg grid cell).**

Yes we agree with the reviewer and indeed we have used a misleading terminology. With "intersections", or "coincidences", indeed we mean the monthly-average number of instrument footprints satisfying the coincidence criteria gridded at 2 x 2 degree resolution. They are computed over a year time span and are averaged over a different time units. E.g. in figure 3 we plot the monthly-average number of instrument footprints satisfying the coincidence criteria gridded at 2x2 degrees.

Figure 5, as the reviewer has noticed, depicts the mean number of ice cloud range bins with a resolution of 500 m per 2 x 2 degree grid cell. This means that a number of 2 corresponds to an area where, on average, there are ice clouds with a vertical thickness of 1000 meters. Then, we multiplied each pixel of the grids reported in Figure 3 for the corresponding pixel of the grids reported in Figure 5 in order to obtain similar grids for calibration points. So, basically, we weight each intersection point on the number of calibrating target we have at that point (e.g. if an intersection occurs where no calibration target is present, that intersection can't be a calibration point; if it occurs where two calibration targets are present, we have two possible points that can be used to calibrate the radar. So, we computed the distribution of calibration points as: (mean number of ice cloud bins per 2 x 2 deg)\*(number of footprints within the 2 x 2 deg grid cell satisfying the coincidence criteria per unit time). We didn't divided for the "(number of footprints per 2 x 2 deg grid cell)", because we are interested in having an absolute number of calibration points. Then, we summed all the elements of the grid containing the number of calibration points, to obtain the total number of calibration points per unit time.

**Lines 130-135: The authors state that due to the low attenuation in ice at Ka and W-band, the measured reflectivity of a given ice cloud target will be nearly identical from near-nadir and large off-nadir viewing geometries. This seems to neglect the possibility of backscatter dependence on angle for oriented ice crystals, which are known to exist in certain stratiform regimes. The potential impact of this effect on the proposed calibration method should be addressed in this manuscript.**

In general the reflectivity may be slightly different in presence of non-spherical particles which are preferentially oriented. However for the calibration we do use reflectivity values that are large, thus corresponding to generally more randomly oriented aggregate. These scatterers tend to have backscattering cross sections independent from incidence angle. With respect to the attenuation the Hitschfeld–Bordan attenuation-correction could be implemented to account for the differential signal between the two viewing directions.

**I find the description of the Z-PDF method at the beginning of Section 2.3 somewhat confusing. My interpretation of the procedure is that a correlated pair of PDFs is calculated for 5-km-long along-track swaths of radar reflectivity profiles that are separated by Delta s, and this pair could be labeled with an index "i", say $f_i(Z)$ and $g_i(Z)$. The JS distances are calculated for these specific PDFs $f_i(Z)$ and $g_i(Z)$ with index i and then an ensemble of such realizations is made by finding N (i = 1, 2, ..., N) pairs of such spatially separated PDFs. In lines 185-188, the single-swath, spatially separated PDFs are defined, but then it is said that the PDFs are accumulated up to a specified sample size before the JS distance is calculated between "the two PDFs". It isn't totally clear what is meant by this accumulation, and could be interpreted as accumulating counts across many spatially separated observation points to form a single PDF consisting of reflectivity observations from N swaths. This is confusing since it seems to be the case that the JS distance can only be meaningfully calculated on isolated pairs of single swath PDFs. I think cleaning up the wording and notation in this section would make the manuscript much more easily readable.**

We agree with the reviewer that the description was confusing. The key of the methodology here presented is to build correlated pair of PDFs which should simulate the PDFs as collected by the calibrating and by the to-be-calibrated radar. For each of the quasi- coincidence criteria we have constructed these PDFs by using points collected by CloudSat or GPM that are separated by a distance $\Delta s$ like illustrated in Fig.4. In each of the 5-km thick layers only a limited number of calibrating points is present. In order to smooth the pdfs a large number of calibrating points is needed. This is achieved by putting together 5-km thick belonging to different orbits. By so doing two PDFs $f_1(Z)$ and $g_1(Z)$ are obtained with a characteristic number of sampling points (higher number of points obviously requiring larger number of orbits). This number of sampling points is increased to assess the impact that the number of calibrating points has in reducing the sampling noise. The exercise is then repeated multiple times (thus involving different orbits or different segments of the same orbit) to find an ensemble of such pair PDFs. We name these realizations $f_i(Z)$ and $g_i(Z)$ with i = 1, 2, ..., N. For each pair, the Janson-Shannon distance $d_{JS}[i]$ is computed. If N is large enough the pdf of all $d_{JS}[i]$ gives an idea about the expected distance between 2 PDFs and its natural variability associated to the fact that in our calibration methdology clouds not from coincident but from quasi-coincident overpasses are sampled.

**Minor comments**

**Line 31: It is not clear why conical scanning permits larger domain coverage, unless the statement is more about the difficulty of reaching the same large incidence angles using a cross track technique vs using a conical scanning approach.**

Yes we agree with the reviewer. In principle conically scanning radars can achieve larger domain coverage because they can reach larger incidence angles, but yes in principle also cross-tracking radars they have the same capability. We will make this clear in the revised version.

**Line 31: The considerations around surface clutter at large off nadir vs near nadir incidence are very different in nature, and it is not a given that one offers lower clutter free heights in general than the other. Specifically, beam width determines the clutter height at large off nadir angles, and the problem is essentially independent of the radar waveform parameters/ the range weighting function. The situation is completely different for near nadir scattering**

where the beam width has little to no impact on the clutter free height, and it is the range weighting function as well as the transmitter phase noise (for high time-bandwidth product PC systems) that determines the clutter. This statement on line 31 should be clarified or removed.

Agreed. We will remove the statement relative to the clutter.

**Line 4 and Line 35: 10 degrees vs 12 degrees for surface scattering-based calibration?** Thanks for pinpointing at this discrepancy. The sweep spot is actually at $11°$, see Tanelli et al CloudSat's Cloud Profiling Radar After Two Years in Orbit: Performance, Calibration, and Processing IEEE TRANSACTIONS ON GEOSCIENCE AND REMOTE SENSING, VOL. 46, NO. 11, NOVEMBER 2008. we will amend this.

**Line 79: More justification is needed for this particular choice of statistical metric, the JS distance, in assessing the similarity between two PDFs. What are the strengths/sensitivities of this metric to PDF structure?**

The Jensen–Shannon distance is a method, commonly used in statistics, to measure similarity between two probability distributions. Given that the base 2 logarithm is used in the definition, it is bounded by 1 with the maximum obtained for non-overlapping distributions which is a clear advantage over the standard L2 distance. Moreover, the JS distance is equal to 0 if and only if two PDFs are equal.

**Line 42: The authors highlight the statistical cross-calibration approach of Protat et al. (2009) that utilizes CFADs to perform a calibration comparison including information about the vertical distribution of observed reflectivity, but the approach implemented in this work collapses the distribution into a 1-D PDF with no vertical information retained. This difference should be addressed with at least some discussion in the manuscript.**

The method from Protat makes some qualitative consideration based on CFADs of reflectivities which could also be repeated in our approach. However when it comes to make quantitative assessment Protat et al., uses the mean vertical profiles of reflectivities (i.e. they condense the 2D distribution in a 1-D PDF by getting rid of the Z-dimension). We decided to get rid of the vertical dimension keeping the dependence on the reflectivities.

**Line 132: The authors state that only ice clouds away from deep convection are used in this analysis, but do not state how the observational data sets have been filtered to ensure such ice cloud detections are excluded. Please include these details.**

We used the CloudLayerType field from the 2B-CLDCLSS CloudSat data product (https://www.cloudsat.cira.colostate.edu/data-products/2b-cldclass) to identify the cloud type, and then exclude the deep convective clouds. Similarly for GPM we have excluded convective pixels. We will include these details in the manuscript.

**Section 2.2: the definition and description of statistical quantities that are intended to quantify the intersection occurrence rate could be improved. Most importantly, the gridding resolution of 2 deg x 2 deg seems rather arbitrary, and for instance, the "mean number of layers" plots in fig would scale (likely proportionally) to this choice of angular area. Furthermore, it seems slightly misleading to refer to this quantity as a number of ice cloud layers, when my interpretation is that this is actually referring to the mean number of radar range bin volumes (e.g., 5 x 5 x .25 km for GPM) that have returns meeting the ice cloud criteria laid out in the paper. It would seem more appropriate to use for this statistical quantity a mean area density of ice cloud bins that would not depend so strongly on the choice of grid resolution. Furthermore, this would allow for a more straightforward calculation of expected mean number of intersection bins per unit time by multiplying this density by the mean inter-satellite swath overlap and the mean rate of overpass "coincidence" for that lat/lon (see major comment 1 above).**

**Assuming my interpretation of the calibrating target definition in Section 2.2 is correct, I suggest that the ice cloud layer variable be renamed to something like "ice cloud radar bins" to ensure that the reader is not confused by this. The term ice cloud layer evokes an extended cloud morphological feature that would nominally encompass a very large number of radar range bin volumes. If my interpretation of this "ice cloud layer" definition is wrong, I suggest that the explanation in this section be reworded for clarity.**

Here the terminology we have used was misleading as well. Indeed what is plotted are ice cloud radar 500-m thick bins. The selection of the $2 \times 2$ gridding is done with the goal of reducing the sampling noise associated to CloudSat and GPM observations. The results are expected to be independent to that resolution because we have normalised the number of detected ice cloud radar 500-m thick bins by the total number of radar footprints at each grid point.

**The vertical black and blue dashed lines in Fig. 4 (right panel) are difficult to distinguish**

We will use a continuous line in correspondence to the black line to make the two lines better distinguishable.

**Section 2.1: More details are needed on how orbits are propagated for the purposes of calculating intersection points**

The orbits have been propagated analytically using the trajectory equation obtained with the integration of the equation of motion for the restricted two-body problem. The initial orbital parameters used in the propagation are reported in Table 2 and 3. The J2 perturbations only have been taken into account.

---

## Author Response (AR2)

**REPLY to REVIEWER 1**

We thank the reviewer for their review and their detailed comments.

All figures have been updated so that the colour schemes used in our maps and charts allow readers with colour vision deficiencies to correctly interpret our findings.

Below you will find the reviewer's comment in bold and our reply.

**L210- 215. When you say "This number of sampling points (which will correspond to a given acquisition calibration period) is increased to assess the impact that the number of calibrating points has in reducing the sampling noise. The exercise is then repeated multiple times (thus involving different orbits or different segments of the same orbit) to find an ensemble of such PDF pairs". It is not clear to me if you also perform a reshuffle of the starting points of the "delta_s" sequence within a given orbit. (To explain me better, look for example at Figure 4 right panel. If in that figure the sequence of the blue and black vertical boxes starts around 8.75 deg Latitude instead of 8.4 deg, you will have a completely different set of calibrating points in that particular case).**

Yes, this is exactly what we have done. Indeed we repeat that procedure also changing the starting points of the "$\Delta s$" sequence within a given orbit. This has been explained in the text

**REPLY to REVIEWER 2**

We thank the reviewer for their review and their detailed comments.

Below you will find the reviewer's comment in bold and our reply.

**I would ask for is a single sentence clarifying the use of the term "ice cloud bins" from the usual understanding of "radar range bin". Specifically, the ice bins in this text are assumed to be horizontally uniform vertical slices that extended throughout the 2 x 2 deg cell, with a depth of 500 m. While a radar range bin has a horizontal extent of (typically) a few kilometers, and a range extent given by the radar range resolution.**

We use the terms "ice cloud bins" or "ice cloud radar range bins" to indicate a (500 m) radar range bin that contains an ice cloud. In Figure 5, in each 2x2deg pixel is shown the mean number of 500m thick ice cloud radar range bins per profile sampled in that pixel. The term "per profile" was missing and should explain the meaning of this quantity. This has now been amended in the text and in all labels of figures. For example, if in a given 2x2deg cell the mean number of 500m thick ice cloud radar range bins per profile is 1, it means that for any vertical profile collected in that region, an average of one 500m thick ice cloud radar range bin will be detected.

**In the new text at the beginning of Section 2.3, there is discussion of 5-km-thick layers that are used for PDF generation, suggesting the that 5-km extent lies in the vertical direction. Isn't the 5 km number referring to the along-track sampling distance?** Yes, thanks for highlighting this imprecision. "5 km" is referred to the along-track sampling distance and the horizontal width of the sampled vertical slice. When referring to this quantity, at the beginning of Section 2.3 we now changed in the text "5-km thick layers" with "5km-wide vertical slices".

---

## Author Response (AR3)

There were no revisions left to implement